# Phased Array Ultrasonic Method for Robotic Preload Measurement in Offshore Wind Turbine Bolted Connections

**DOI:** 10.3390/s24051421

**Published:** 2024-02-22

**Authors:** Yashar Javadi, Brandon Mills, Charles MacLeod, David Lines, Farhad Abad, Saeid Lotfian, Ali Mehmanparast, Gareth Pierce, Feargal Brennan, Anthony Gachagan, Carmelo Mineo

**Affiliations:** 1Centre for Ultrasonic Engineering (CUE), Department of Electronic & Electrical Engineering (EEE), University of Strathclyde, Glasgow G1 1XQ, UK; brandon.mills@strath.ac.uk (B.M.); charles.macleod@strath.ac.uk (C.M.); david.lines@strath.ac.uk (D.L.); s.g.pierce@strath.ac.uk (G.P.); a.gachagan@strath.ac.uk (A.G.); 2Department of Design, Manufacturing & Engineering Management (DMEM), University of Strathclyde, Glasgow G1 1XQ, UK; 3Department of Naval Architecture, Ocean & Marine Engineering (NAOME), University of Strathclyde, Glasgow G1 1XQ, UK; farhad.abad@strath.ac.uk (F.A.); saeid.lotfian@strath.ac.uk (S.L.); ali.mehmanparast@strath.ac.uk (A.M.); feargal.brennan@strath.ac.uk (F.B.); 4Institute for High Performance Computing and Networking, National Research Council, Via Ugo La Malfa 153, 90146 Palermo, Italy; carmelo.mineo@icar.cnr.it

**Keywords:** ultrasonic stress measurement, phased array ultrasonic testing (PAUT), offshore wind turbines (OWT), total focusing method (TFM), robotics, non-destructive testing (NDT)

## Abstract

This paper presents a novel approach for preload measurement of bolted connections, specifically tailored for offshore wind applications. The proposed method combines robotics, Phased Array Ultrasonic Testing (PAUT), nonlinear acoustoelasticity, and Finite Element Analysis (FEA). Acceptable defects, below a pre-defined size, are shown to have an impact on preload measurement, and therefore conducting simultaneous defect detection and preload measurement is discussed in this paper. The study demonstrates that even slight changes in the orientation of the ultrasonic transducer, the non-automated approach, can introduce a significant error of up to 140 MPa in bolt stress measurement and therefore a robotic approach is employed to achieve consistent and accurate measurements. Additionally, the study emphasises the significance of considering average preload for comparison with ultrasonic data, which is achieved through FEA simulations. The advantages of the proposed robotic PAUT method over single-element approaches are discussed, including the incorporation of nonlinearity, simultaneous defect detection and stress measurement, hardware and software adaptability, and notably, a substantial improvement in measurement accuracy. Based on the findings, the paper strongly recommends the adoption of the robotic PAUT approach for preload measurement, whilst acknowledging the required investment in hardware, software, and skilled personnel.

## 1. Introduction

Bolted connections play a vital role in various industries including manufacturing and defence, as well as critical infrastructure such as Offshore Wind Turbines (OWT), aerospace, vehicles, ships, railways, bridges, and buildings. Each application has specific construction standards, such as BS EN 14399 [1] and ASME PCC-1 [2], designed to ensure reliable and durable bolted connections that meet structural integrity and stability requirements [3]. Bolted flange connections, crucial mechanical joints in wind turbine support structures, require improvement especially when subjected to harsh marine environments [4,5,6]. Monitoring the preload and stress of bolts is essential to prevent loosening and connection failures. However, the current inspection method in the offshore renewable energy sector involves frequent checks by inspectors, contributing to the overall injury rate of three times higher than that in offshore oil and gas applications [7,8,9]. The current standard inspection procedure for bolt testing involves fixed permanent strain gauges [10] and/or the use of single-element ultrasonic transducers [3].

When using strain gauges for industrial applications, the main challenge lies in the large number of bolts used in wind turbines, resulting in the need for thousands of strain gauges. Consequently, only a limited number of bolts can be monitored as a common practice [11]. Some commercially available washer-shaped strain gauges [12], which are also utilised in this paper, can facilitate bolt preload measurement, although such systems are primarily used for research and development purposes in the laboratory due to the high cost of installation for all bolts and logistical challenges associated with data acquisition. To address this issue, this paper investigates a robotic ultrasonic method, which allows discrete monitoring of multiple bolts with a single permanently available robot inside the OWT. This approach minimises human involvement and enables comprehensive and continuous monitoring of bolts [4,5,8].

The ultrasonic preload measurement technology relies on the theory of acoustoelasticity, which establishes a relationship between acoustic wave velocity and material stress, as well as the change in ultrasonic Time of Flight (ToF) corresponding to the change in bolt length resulting from the axial tightening force [13]. The calibration procedure, including the measurement of the acoustoelastic coefficient and ToF in the stress-free bolt, is essential for this process [3]. Commercially available equipment can employ this approach for stress measurement, which uses single-element ultrasonic transducers. The current industry standard for ultrasonic inspection of bolts utilises single-element transducers and assumes that any difference between the ToF of a bolt in service and the calibration bolt corresponds to a change in stress (preload). While this assumption may hold true for brand-new bolts, similar to those used in the laboratory for calibration, it overlooks factors such as corrosion, defects, ageing, creep, strain-hardening, fatigue, and other material changes that occur during the service life. In this paper, the innovative use of the Phased Array Ultrasonic Testing (PAUT) system is proposed as an alternative to the single-element approach. The advantage of the PAUT system over single-element transducers lies in its capability for defect detection and simultaneous stress measurement.

PAUT systems are commonly employed for defect detection [11] and are preferred over single-element transducers due to their general advantages such as wider scanning areas, focusing capability, higher inspection quality, flexibility, and shorter inspection time resulting from rapid visualization [14]. By employing a PAUT system instead of a single-element transducer, it becomes possible to identify potential defects within specific acoustic paths utilised for ToF measurements and subsequently utilise alternative acoustic paths for stress measurement. Additionally, phased-array probes enable various inspections to be conducted from a single location through synthetic aperture focusing and the ability to steer the ultrasonic beam across different angles and positions [15,16,17]. The development of 2D phased-array ultrasonic imaging transducers has also expanded the application of 3D volumetric imaging of components [18], which is particularly beneficial for bolt inspections. Furthermore, advanced post-processing algorithms like the Total Focusing Method (TFM) have made it possible to focus on bolt threads, which are critical areas of concern in OWT bolts. The TFM, employing a delay-and-sum amplitude method with synthetic focusing at pixels in the discretised imaging domain, provides a clearer scanning image of threads and their defects. The high-alloy steel typically used for offshore applications in OWT bolts results in a poor Signal-to-Noise Ratio (SNR), making Phase Coherence Imaging (PCI) another advantage of PAUT. PCI is an amplitude-free synthetic beamforming method that considers phase dispersion at each discrete image point, effectively reducing incoherent noise caused by side lobes, grating lobes, reverberations, and grain noise [19].

Despite the numerous advantages of PAUT over single-element transducers in the area of defect detection, its application for bolt preload measurement is considered in this paper for the first time. Therefore, the main novelty of this paper lies in bolt preload measurement using the PAUT method. This opens the door to new developments, such as robotic preload measurement and considering the effect of defects on stress measurement, thanks to the comprehensive PAUT defect detection (along with the addition of advanced post-processing algorithms). It is worth mentioning that the benefits of using PAUT systems to measure stress (mainly residual stress) in other applications, like welding, were demonstrated by Javadi et al. [20]. 

This study presents an innovative exploration of the application of robotic PAUT in the field of bolt testing. The primary objective is to enhance maintenance efficacy and facilitate the early detection of potential faults. By accurately analysing preload characteristics during maintenance procedures, the identification of latent imperfections within the fastener system becomes a feasible proposition. The utilisation of robotic systems in this context provides a twofold advantage. It significantly diminishes the likelihood of procedural errors, concurrently augmenting measurement safety. This paper serves a dual focus, concurrently addressing robotics, defect detection, and bolt preload measurement through the PAUT method. This comprehensive approach marks a pioneering endeavour within this relatively unexplored domain. 

## 2. Theoretical Background

Ultrasonic velocity and stress can be related to the material’s elastic properties. The relationship between these two factors can be described using Equation (1):(1)$$V=\sqrt{\frac{V_{0}+K\sigma }{\rho }}$$

Here, *V* represents the ultrasonic velocity in the material, *V*_0_ is the intrinsic or unperturbed velocity of sound in the material, *K* is the pressure derivative of the velocity with respect to stress (sometimes referred to as the velocity–stress constant), *σ* is the stress applied to the material, and *ρ* is the density of the material.

Equation (1) provides a simplified representation of the relationship between stress and velocity, disregarding the complex behaviour of materials under stress. However, acoustoelasticity focuses on the nonlinear relationship between stress and ultrasonic velocity. In acoustoelasticity, the nonlinear relationship between stress and ultrasonic velocity is typically represented by higher-order polynomial equations or empirical models. These models consider the nonlinear effects that arise when stress levels become significant. One commonly used nonlinear relationship in acoustoelasticity is the third-order polynomial equation, often referred to as the acoustoelastic equation. It can be expressed as Equation (2):(2)$$\frac{\Delta V}{V_{0}}=A\sigma +B\sigma^{2}+C\sigma^{3}$$
where Δ*V*/*V*_0_ represents the fractional change in ultrasonic velocity due to stress, *A*, *B*, and *C* are coefficients that depend on the material properties and can be determined through experimental calibration, and *σ* denotes the applied stress. This suggests that the change in ultrasonic velocity is not linearly proportional to the stress but instead involves higher-order terms such as the stress squared (*σ*^2^) and cubed (*σ*^3^). The coefficients *A*, *B*, and *C* determine the magnitude of these nonlinear effects and vary depending on the specific material being studied. By measuring the change in ultrasonic velocity under different stress conditions and fitting the data to Equation (2), it is possible to determine the coefficients *A*, *B*, and *C* for the material, which measures the acoustoelastic coefficients and acts as a calibration procedure.

Hughes and Kelly [21] were the first to publish the analytical expressions that describe the changes in bulk wave velocity within a pre-stressed isotropic solid. These expressions were developed using Murnaghan’s theory of finite deformations of solids [22]. In the context of acoustoelasticity, the nonlinear effects are not specifically related to Lame’s constants or the Murnaghan equation of state. Lame’s constants (*λ* and *μ*) are parameters used to describe the elastic behaviour of isotropic materials in linear elasticity theory and determine the relationship between stress and strain in linear elasticity. The Murnaghan equation of state is an equation that relates the pressure, volume, and bulk modulus of a material under hydrostatic compression. While the Murnaghan equation of state can describe the nonlinear behaviour of materials under pressure, it is not specifically related to the nonlinear effects observed in acoustoelasticity, which involve the relationship between stress and ultrasonic velocity.

In ultrasonic stress measurement for bolts, two common formulas can be used: one based on a linear relationship and another based on a nonlinear relationship. The formula based on a linear relationship assumes that the stress-induced change in ultrasonic velocity is directly proportional to the applied stress. The formula for measuring stress using this linear relationship is represented by Equation (3):(3)$$\sigma =\frac{\left(\Delta V/V_{0}\right)}{A}$$

Here, *A* represents the velocity–stress constant specific to the tested material and bolt.

The nonlinear relationship takes into account the observed nonlinear effects in acoustoelasticity and provides a more accurate estimation of stress, particularly at higher stress levels. Equation (2) can be applied for bolt preload measurement based on the assumption of nonlinearity. In practical applications, both linear and nonlinear approaches can be utilised for ultrasonic stress measurement in bolts. The linear relationship offers a quick estimation of stress, while the nonlinear relationship provides higher accuracy, especially when dealing with significant nonlinear effects at higher stress levels. Despite some literature simplifying the bolt stress measurement by assuming a linear relationship (Pan et al., 2020 [22]), this paper employs the nonlinear relationship. This decision is driven by the objective of enhancing the accuracy of the ultrasonic method using the innovative PAUT approach, requiring the use of the best approach (with minimal simplification assumptions) for single-element measurement.

Equation (2) is rearranged in terms of the Time-of-Flight (ToF) measurement, as shown in Equation (4):(4)$$\sigma =\frac{1}{2A}\left[\frac{\Delta t}{t_{0}}-B{\left(\frac{\Delta t}{t_{0}}\right)}^{2}-C{\left(\frac{\Delta t}{t_{0}}\right)}^{3}\right]$$

In Equation (4), Δ*t* denotes the change in the Time-of-Flight (ToF) difference due to the applied stress and *t*_0_ represents the baseline or unperturbed time of flight. Coefficients *A*, *B*, and *C* are the same material-dependent coefficients described in Equation (2), and they are specific to the material of the bolt being tested. To minimize the impact of varying couplant layers, it is an industry standard to consider the difference between the second backwall echo and first backwall echo as the Time of Flight (ToF). Consequently, *t*_0_ and Δ*t* in Equation (4) are calculated using Equations (5) and (6), respectively:(5)$$t_{0}=t_{2-0}-t_{1-0}$$
(6)$$\Delta t=\left(t_{2-1}-t_{1-1}\right)-(t_{2-0}-t_{1-0})$$
where *t*_2−0_ represents the ToF of the second backwall echo in the stress-free material, *t*_1−0_ represents the ToF of the first backwall echo in the stress-free material, *t*_2−1_ represents the ToF of the second backwall echo in the stressed material, and *t*_1–1_ represents the ToF of the first backwall echo in the stressed material.

Temperature is another crucial factor to consider in ultrasonic stress measurement. The speed of sound in a material is influenced by temperature, and temperature variations can impact the accuracy of ultrasonic velocity measurements. To ensure precise and reliable results, it is necessary to take into account the temperature difference between the calibration and testing conditions. In this paper, both the calibration and primary robotic tests were conducted in a temperature-controlled robotic laboratory to minimize the influence of temperature. However, since the concept of robotic PAUT stress measurement is proposed for more accurate in situ measurements, it is essential to record the temperature during the calibration process and the in situ tests. Equation (7) should be used for stress measurement, considering temperature:(7)$$\sigma =\frac{1}{2A\beta }\left[\left(E\frac{\Delta t}{t_{0}}-\alpha \Delta T\right)-B{\left(E\frac{\Delta t}{t_{0}}-\alpha \Delta T\right)}^{2}-C{\left(E\frac{\Delta t}{t_{0}}-\alpha \Delta T\right)}^{3}\right]$$

Here, *E* is the elastic modulus and both Δ*t* and *t*_0_ are given by Equations (5) and (6). The coefficient *α* is used to account for the temperature difference between the calibration and testing conditions (obtained through calibration at different temperatures), and Δ*T* represents the temperature difference. Additionally, the coefficient *β* is included in Equation (7) to consider the couplant gel thickness, although, in most industry-practice non-destructive evaluation (NDE) approaches, compensating for this effect is achieved by measuring the difference between the second and first echoes, from Equations (5) and (6).

Now, Equation (7) needs to be adapted to the PAUT approach:(8)$$\sigma =\frac{1}{2An\beta }\overset{n}{\underset{i=1}{\sum }}\left[\left(E\frac{\Delta t_{i}}{t_{i0}}-\alpha \Delta T\right)-B{\left(E\frac{\Delta t_{i}}{t_{i0}}-\alpha \Delta T\right)}^{2}-C{\left(E\frac{\Delta t_{i}}{t_{i0}}-\alpha \Delta T\right)}^{3}\right]$$

In Equation (8), *t_i_*_0_ represents the ToF corresponding to the wave sent by element *i* and received by the same element. The variable *n* denotes the number of healthy A-scans, which refers to those that are not obstructed by potential defects. Javadi et al. [23] referred to this approach as the PAUT Direct Approach in residual stress measurement. It bears similarity to bolt testing, with the distinction that in residual stress measurement, *n* represents the number of elements, whereas, in bolt testing, not all acoustic paths can be utilised due to potential blockages by defects.

The alternative PAUT approach is referred to in the above paper as the PAUT-FMC Approach where FMC stands for Full Matrix Capturing. This method takes into account all possible combinations in the Full Matrix Capturing technique, where each acoustic path can be generated by element *i* and received by element *j*, as illustrated in Equation (9):(9)$$\sigma =\frac{1}{2An^{2}\beta }\overset{n}{\underset{i=1}{\sum }}\overset{n}{\underset{j=1}{\sum }}\left[\left(E\frac{\Delta t_{i/j}}{t_{i/j0}}-\alpha \Delta T\right)-B{\left(E\frac{\Delta t_{i/j}}{t_{i/j0}}-\alpha \Delta T\right)}^{2}\;-C{\left(E\frac{\Delta t_{i/j}}{t_{i/j0}}-\alpha \Delta T\right)}^{3}\right]$$

In this paper, Equation (9) and the PAUT-FMC approach are not utilised, and are included only for interest. Instead, the selection of healthy A-scans is based solely on the direct method, involving the sending and receiving of signals with the same element. This choice is made due to the complexities associated with stress measurement in materials containing known defects, such as bolts in the context of this paper. The presence of defects introduces the possibility of oriented acoustic paths that involve different sender and receiver elements passing through the defect, which can influence the resulting acoustic path. Therefore, Equation (8) is employed in this paper to calculate the stress in bolts.

## 3. Methodology and Experimental Setup

### 3.1. Methodology

The integration of defect detection and preload measurement is of the utmost importance for the life assessment of engineering assets. Failure to do so can lead to potential influences on the ultrasonic stress measurement and calibration procedure due to the presence of defects. In this paper, and especially with the proposed robotic PAUT approach, additional measurements can be made to provide a more comprehensive compensation. Specifically, the robot’s Z position can be measured accurately, enabling the precise measurement of the couplant gel thickness as a complement to the analytical compensation provided in Equation (7). Furthermore, the robot is equipped with a load cell (force/torque sensor) that ensures constant pressure on the transducer, resulting in a consistent couplant gel thickness. The implementation of advanced post-processing algorithms such as the TFM allows for improved focusing capabilities on critical areas, such as threads, which are typically of significant concern in safety-critical bolts. The methodology used for the robotic PAUT of the bolt is shown in Figure 1 and explained below:(1)A sector scan is performed to simultaneously inspect the bolt threads and the internal volume for defects, and the acceptance criteria are evaluated to determine if the defects meet the requirements. The acceptance criteria, based on industry performance standards from ASME codes (B16.5 [24], B31.3 [25], and PCC-1 [2]), state that the defect size should be within 10% of the nominal bolt size. In this paper, the bolt under investigation is M36, and small defects are defined as having a size of less than 3.6 mm.(2)Even if the bolt is rejected due to larger defects exceeding 3.6 mm, the automation process discussed in this paper remains valuable as it enables automatic replacement of the bolt. However, this automatic replacement is beyond the scope of this paper and will not be discussed.(3)This paper primarily focuses on small defects, as it is believed that they can still affect ultrasonic stress measurements.(4)To ensure small defects meet the acceptance criteria, the unique advantages of PAUT over single-element transducers (such as TFM, PCI and Focused B-Scan) are employed for comprehensive investigations. This is crucial to mitigate potential misinterpretations of defect size caused by low SNR and other inspection challenges encountered during in situ testing of OWTs. If a large defect is detected at this stage, the procedure described in Point 2 will be repeated.(5)If the defect is deemed acceptable after this in-depth study, a 3D volumetric scanning image is generated using the PAUT system.(6)The PAUT probe position is adjusted by a robotic system based on the 3D image of the bolt defects, aiming to minimize the interference of small defects with the ultrasonic wave propagation inside the bolt. This adjustment process is referred to as hardware adjustment in this paper.(7)In cases where complete hardware adjustment is not feasible, meaning that some defects still obstruct the acoustic path, software adjustment is implemented. Since the ultrasonic array can generate multiple acoustic paths, only the acoustic paths free from defects obstructing the backwall are considered in the next stages. These selected paths are termed “healthy A-scans” (see Figure 2).(8)Time-of-Flight (ToF) measurements required for stress calculation are exclusively conducted on the healthy A-scans.(9)The final step involves post-processing and utilising acoustoelasticity for stress calculations.

As depicted in Figure 1, FEA is employed in this paper to provide support for the ultrasonic measurement outcomes. The ultrasonic method enables the measurement of stress averages [26,27] within the stress domain located along the acoustic path. For the bolt, the primary stress domain lies between the bolt head and nut, specifically in the flange connection area. Conversely, the region between the nut and the bolt’s free side typically experiences minimal stress. Due to the placement of the ultrasonic probe on the free side of the bolt, with the bolt head considered as the backwall, the acoustic path encompasses both the stress-free area and the flange areas of the bolt. Consequently, the stress measured by the ultrasonic probe represents the average of zero stress in the stress-free region and high stress in the tensioned flanged area. This paper utilizes FEA to analyse the average stress, allowing for comparison with the ultrasonic measurements.

### 3.2. The Influence of Small Defects on the Stress

To investigate the impact of small defects that meet the acceptance criteria, an M36 bolt (made of A4-70 stainless steel) was subjected to testing using a 2.25 MHz array from the bottom (Figure 3). The bolt head featured engraved text with a depth of 0.5 mm. These engraved marks are reflected in the backwall echo detected by the ultrasonic waves. In the first test, the bolt was tested under normal conditions (Test #1). However, in the second test, the engraved marks were filled with ultrasonic gel to observe their effect on the ultrasonic echo (Test #2).

### 3.3. Experimental Setup for Robotic PAUT of Bolt

The experimental setup for the robotic PAUT of the bolt is illustrated in Figure 4. The setup comprised an M36 bolt subjected to testing using a 2.25 MHz, 20-element array (Sonatest Ltd., Milton Keynes, UK) with a pitch of 1.2 mm and an elevation of 12 mm. A PEAK LTPA with 32:64 channel inputs and high dynamic range functionality (PEAK NDT, Derby, UK) was used as the phased-array controller. To validate the ultrasonic stress-measurement results, a washer-shaped load cell (BoltSafe, Beuningen, The Netherlands) was employed. The probe was connected to the flange of a 6-axis robot (KUKA Robotics, Augsburg, Germany) using a custom 3D-printed mounting. Additionally, the robot was equipped with a Force/Torque (F/T) sensor, which played a critical role in maintaining consistent pressure on the ultrasonic probe, ensuring improved measurement repeatability. Steel plates were utilised to form the flanged stressed area between the nut and the bolt head. Although the bolt head is not visible in the image as it rests on the robot table beneath the bottom steel plate, the black spanner is attached to it which indicates its position within the setup. The tightening process was performed manually using the spanners. A LabView code was utilised for the PAUT data acquisition and integration of all four key systems: the robot, F/T sensor, washer-shaped load cell, and PAUT system.

## 4. Results and Discussions

### 4.1. The Influence of Small Defects on the Stress and Advantages of the PAUT System

The differences between the A-scans of Test#1 and Test#2 are illustrated in Figure 5, with a focus on the orange plot. Notably, the zero crossing has shifted from 62.48 µs to 62.515 µs, indicating a difference of 35 nanoseconds. It is important to recognize that every 10 nanoseconds corresponds to a 40 MPa discrepancy [28,29]. Therefore, the observed 35 ns difference implies a potential error of up to 140 MPa. However, it is crucial to exercise caution and acknowledge that part of this error may be attributed to variations in the thickness of the couplant layer. To account for this, the disparity between the first and second backwall is taken into consideration, as depicted in Figure 6 and Table 1.

From Table 1, a variation of 10 nanoseconds in the Δ*t* between Test#1 and Test#2 is evident. This difference corresponds to a 40 MPa error, not accounting for any errors caused by variations in the couplant film. It is important to note that this substantial error occurred solely due to the presence of gel in the bolt stamp, while both tests had zero stress. It is worth considering that during the service life of a bolt used in the harsh conditions of an offshore wind turbine, a much thicker layer of corrosion, dust, grease, and other field pollutants can be expected. In a hypothetical scenario, the utilisation of a single-element transducer for stress measurement and conducting calibrations in a corrosion-free laboratory environment raises significant challenges when deploying an operator to the field. Specifically, the operator may encounter difficulty in identifying the existence of a 0.5 mm corrosion layer. As a result, there is a risk that the operator may misinterpret the observed 10-nanosecond discrepancy as a variation in bolt stress. Regrettably, such a misinterpretation could lead the asset owner to erroneously perceive a 40 MPa alteration in bolt tension, consequently incurring unnecessary costs associated with inspection, replacement, and repairs.

This circumstance underscores the importance of employing the PAUT system, due to the above-mentioned capabilities for effectively identifying the presence of a corrosion layer. By employing hardware and software adjustments, as illustrated in Figure 1, the impact of this defect on subsequent stress-measurement procedures can be minimized. Adopting this approach ensures higher accuracy and reliability in evaluating bolt integrity, enabling more informed decision-making while optimising resource allocation for maintenance activities.

The consideration of small defects warrants attention in the context of bolt-replacement practices. While some industries may advocate for the immediate replacement of a bolt upon detecting any defect, regardless of its size, with the subsequent exclusion of stress measurement, it is essential to acknowledge the cost implications associated with such unnecessary repair and replacement actions. The determination of what qualifies as “unnecessary” is dictated by established standards and codes. In this study, the acceptance criteria of 3.6 mm, derived from a real-life case study, aligns with the performance standard developed by one of the asset operating companies based on ASME codes. Consequently, if a defect falls below the 3.6 mm threshold, there is no requirement to reject or replace the bolt. Moreover, our investigation has demonstrated that even a significantly smaller defect, such as a 0.5 mm engraved text (which accounts for only 14% of the acceptable defect size), can lead to a substantial 40 MPa measurement error in stress. Therefore, it is not prudent to advocate for bolt replacement when the detected defect surpasses the acceptance criteria by a considerable margin.

### 4.2. The Benefit of Robotics for Preload Measurement in OWT Bolted Connections

It can be argued that this methodology could be performed manually, as PAUT has been consistently used by defect-detection operators in the field. While operator safety, especially in harsh conditions such as offshore wind energy, is the main justification for the necessity of automation, manual inspection presents various technical challenges. These challenges include the lack of control over accurate probe positioning, inconsistent pressure applied to the probe, and variations in the thickness of the couplant layer.

To investigate these problems, a manual test was conducted three times on a healthy bolt, and the results are presented in Table 2. In Test #A, normal pressure was applied to the probe, while in Test #B, the hand pressure was significantly increased. Interestingly, the results were similar, which can be attributed to the idea of measuring the difference between the first and second backwall echoes (Δ*t*) to mitigate the couplant effect. However, Test #C presented different results, as changing the orientation of the probe led to 20 ns variations, equivalent to a substantial 80 MPa. By orientation change, we refer to θz, as depicted in Figure 7. Since the tested bolt was healthy, such a change in ToF could be attributed to material texture and grain orientation, as these bolts are cold-worked. Regardless of the reason, which is outside the scope of this paper, this must be regarded as a significant drawback when conducting manual ultrasonic stress measurements. Alternatively, if both laboratory and field tests are performed robotically, as proposed in this paper, it becomes possible to precisely replicate the orientation and position at which the calibration data were recorded in the lab.

### 4.3. Robotic PAUT of Bolt

#### 4.3.1. Sector Scan

A defective bolt was subjected to testing using the robotic PAUT setup illustrated in Figure 4. The bolt contained a single Side Drilled Hole (SDH) measuring 3 mm in diameter and with a depth of 10 mm. According to the acceptance criteria employed in this paper (3.6 mm defect size), the bolt was deemed acceptable. Two sector scans were conducted, with the probe positioned at 0- and 90-degree orientations (as indicated by θz in Figure 7). This practice is commonly employed by PAUT operators when testing bolts. The defect was not detected on the scan following the above practice, and so the further investigation procedure was initiated (refer to Figure 8). This entailed consideration of additional scanning positions, with the robotic scanner rotating in five-degree intervals as opposed to solely at 0- and 90-degree orientations. Additionally, advanced PAUT techniques, such as the TFM and PCI, were used. These will be discussed in greater detail in the following section (Section 4.3.2). Based on the flowchart depicted in Figure 1, the procedure could have been halted at this point if large defects were detected during the sector scan, resulting in the rejection of the bolt. In such cases, further investigation and subsequent stress measurement would be deemed unnecessary. However, in this paper this scenario did not occur as no large defects were detected using the sector scan, which typically relies on the common practice of scanning at 0- and 90-degree orientations.

It is worth mentioning that SDHs smaller than the critical size (3.6 mm) were intentionally manufactured for a specific purpose. The main question being addressed is whether these small defects, which might be considered acceptable or potentially missed by inspectors, especially when using single-element transducers, can still have an impact on preload measurement. Larger defects are generally easier to detect, even with single-element transducers, without utilising sophisticated PAUT and TFM systems. Therefore, manufacturing large defects would not present a significant challenge, as the bolt would be readily rejected and replaced (see Figure 1).

#### 4.3.2. Further Investigations (Focused B-Scan, TFM and PCI)

Subsequent investigations were conducted since no defect was detected during the sector scan. These investigations involved performing a focused B-scan, although the exact focus range was unknown due to the blind nature of the test. Additionally, the TFM and PCI were utilised: the TFM addressed focusing-range uncertainty, while PCI tackled the low SNR issue discussed in Section 1. Therefore, TFM and PCI techniques were employed, along with utilising the robot to repeat the test at various positions in the X, Y, and θz directions. Two TFM images are presented in Figure 9, where one (Figure 9a) does not reveal the defect, while the other (Figure 9b) clearly displays its presence. It should be noted that the physical position of the probe differed between these figures. Nevertheless, the TFM technique proved instrumental in detecting these relatively small defects. It is important to highlight the limited depth of the defect, particularly in comparison to the bolt threads, which necessitates effective image differentiation to ensure that the reflection from the defect is not masked by signals originating from the threads. After sizing the defect through comparison to a PAUT calibration block with side-drilled holes, the defect was deemed acceptable, giving us the green light to proceed to stress measurement.

It is important to note that the necessity of employing the TFM and PCI should be determined on a case-by-case basis. In this study concurrent data collection, by normal B-Scan and FMC, was conducted using a robotic system. LabView and Matlab programming were employed for sequential TFM and PCI, specifically for processing the FMC data. Referring to the flowchart depicted in Figure 1, if a large defect is detected by the live B-Scan, the bolt will be rejected, rendering subsequent TFM and PCI testing unnecessary. Conversely, in cases where the live PAUT image fails to detect any significant defects, a cautious approach is proposed, involving more comprehensive post-processing of the data to ensure that no defects have been overlooked by the initial B-Scan.

The order the TFM and PCI are implemented in is flexible and dependent on the software used for the test. In the system utilised for this study, it was possible to visualise live TFM results thanks to modifications made to the LabView for simultaneous data collection and fast post-processing. However, PCI was carried out in post-processing, and so in this case the TFM was performed prior to PCI. Nevertheless, this order is not critical; the primary goal is to detect all defects and compare them against the acceptance criteria.

In terms of the array’s positioning, θz was adjusted several times, with a five-degree increment each time, using the robot to encompass various vertical cross-sections of the bolt. On each occasion, FMC data were collected for TFM and subsequent PCI post-processing. Considering the array’s size and the bolt’s dimensions, which are quite well-matched in this work, there was no need to change both X-Y and θz positions, as the variations in θz adequately covered the entire bolt range. However, it is essential to emphasise that this decision depends on the specific case. If the array’s size is significantly smaller than the bolt, then adjusting both X-Y and θz positions may be necessary to cover all sections of the bolt. While this may appear time-consuming, it is worth noting that the robotic testing approach proposed in this paper can offer substantial benefits. 

#### 4.3.3. Hardware and Software Adjustment for Stress Measurement

The hardware adjustment refers to undoing the actions performed in the previous section to detect the defect, achieved by changing the position of the robot. Ideally, ToF measurements should be conducted in the region of the bolt that is defect-free. While this was a straightforward task for the specific case examined in this paper, it may prove impractical for certain types of defects. Utilising PAUT enables us to perform software adjustments as well, which involve identifying the A-scans that do not include the defect. An illustration of this process is presented in Figure 10, where the LabView code is modified to alter the position of the acoustic path, namely Acoustic Paths 1, 2, and 3, generating different A-scans. The healthy A-scan corresponds to Acoustic Path 1 (Figure 10a), which excludes the defect. However, reflections from the defect can still be observed, with Acoustic Path 2 passing through the centre of the defect exhibiting the strongest reflection, while Acoustic Path 3, passing along the edge of the defect, shows a weaker reflection. It is advisable to exclude these reflections from the ToF and stress measurements to minimize error.

In this paper, the hardware and software adjustment process required manual intervention. This was necessary as the PAUT images had to be interpreted, and, subsequently, a suitable position (X, Y, and θz) with no defects in the acoustic path had to be selected by the operator. Following this, the robot’s end effector (the PAUT probe) was positioned correctly for stress measurement. The possibility exists to incorporate Machine Learning-enabled technologies for the scanning image processing and linking this interpretation to the subsequent robot positioning. Such automation could enable the entire process to be executed without human intervention. However, it is essential to note that the development of such a system represents the next phase of this research project and lies beyond the scope of this paper.

In Figure 10, the position of the defect is far from the backwall, making it relatively easy to differentiate between the reflection of the defect and the backwall echo. However, this task becomes significantly more challenging for defects located close to the backwall, as the reflection from the defect may be masked within the backwall echo. In such cases, stress measurement can also be affected, as it becomes unclear whether the peak amplitude represents the defect or the backwall echo. It is important to note that this differentiation is in the defect-detection section and must be conducted before determining the healthy A-scans and proceeding with preload measurement (see Figure 1). This presents one of the anticipated challenges in bolt inspection, necessitating further investigation using the focused B-scan and TFM to ensure careful examination of abnormal signals. The PAUT approach, in comparison with the single-element system, can facilitate detecting such abnormalities, as a comparison between A-scans can help identify anomalies, including reflections from defects very close to the bottom surface.

#### 4.3.4. Finite Element Analysis

In the experimental setup utilised in this paper, a washer-shaped load cell was employed to measure the applied load and, consequently, the stress. However, a direct comparison between the stress measured by the ultrasonic method and the load cell readings is not possible because the ultrasonic method only provides an average stress measurement. To address this issue, FEA was employed to calculate the average stress for comparison with the ultrasonic results. 

The bolt and nut were modelled as 3D solids using C3D8R elements for meshing in the computational analysis. To simplify the finite element simulation and focus on obtaining the average stress along the central path, the impact of threads was not included. The fixed boundary condition was applied to limit the degrees of freedom on the contact surfaces of the bolt head and nut interacting with the plate and washer. Following our experimental setup, the nut was positioned at a specific distance from the bolt head, and tie contact was established between the surfaces where the bolt and nut interact. A bolt load equivalent to the load cell data was applied to generate stress along the central path. Also, the defect in the bolt was considered in the model during the stress analysis process to evaluate its influence on stress analysis paths. 

Figure 11 displays the FEA mesh and corresponding results. It is worth noting that the position of the nut in Figure 11a accurately reflects its initial position in the experimental setup. The average stress along the central path (highlighted in Figure 11b) was calculated for a specific load recorded by the washer-shaped load cell. As an illustration, Figure 11c depicts the stress distribution for a load cell measurement of 30 kN. It is observable that the primary stress area is situated between the nut and bolt head. However, it is important to note that the ultrasonic method measures average stress, encompassing all points along the central path, including some with zero stress and others with maxi-mum stress. 

The averaging effect is demonstrated in Figure 12, which presents multiple load cell increments used as input for the finite element (FE) model in line with the robotic PAUT experiment. For instance, when the load cell indicates 30 KN, the maximum stress is 30 MPa, while the average stress is only 10.89 MPa. Therefore, it is crucial to consider the average stress of 10.89 MPa when verifying the ultrasonic results. Consequently, the approach of simultaneous FEA and ultrasonic stress measurement is highly recommended.

#### 4.3.5. Stress Measurement Using the PAUT Method

The robotic phased array ultrasonic testing (PAUT) experiment was conducted following the methodology outlined in Figure 1. The procedure involved conducting a sector scan, followed by the total focusing method (TFM) for defect detection and subsequent adjustments in both hardware and software settings. The optimal position of the robot was determined through hardware adjustments, while the selection of healthy A-scans was achieved through software adjustments. To gather data at each load increment, the bolt was incrementally loaded and unloaded. This resulted in a significant amount of data, as multiple forces were applied during both the loading and unloading stages. Furthermore, data acquisition was repeated multiple times at each increment, including attaching and detaching the robot, to assess the repeatability of measurements.

The calibration coefficients (A, B, and C) in the third-order polynomial and the nonlinear acoustoelasticity equation (Equation (8)) were determined through numerical methods by utilising the PAUT measurement data as input. The Newton–Raphson method is a commonly used approach to solve these equations, but due to its iterative nature, a numerical solver such as the SciPy library in Python can be advantageous. However, in this study, a large number of data points were available, enabling an alternative numerical approach to be employed. This section focuses on comparing the numerical methods for two approaches: manual scanning with a single-element transducer, and using a robotic phased array with the FMC approach as described in Equation (9). The aim is to investigate the advantages of having a larger dataset in each case and its impact on estimating the acoustoelastic coefficients (A, B, and C).

(a)Single-Element Transducer (manual scanning): With a single-element transducer, only a single set of data points for Time of Flight (ToF) and stress measurements can be obtained. As the dataset is limited, the approach involves directly solving the equations for A, B, and C using numerical methods such as Newton–Raphson or optimization algorithms. The single-element transducer approach offers simplicity and ease of implementation. It requires minimal data processing and computational resources. However, its accuracy and reliability may be limited due to the small dataset and potential uncertainties associated with manual scanning.(b)Robotic Phased Array (FMC approach): Utilising a robotic phased array with FMC enables the acquisition of a significantly larger dataset. Compared to the single-element transducer case, a 20-element array, as utilised in this study, can generate a dataset that is 400 times larger (20 × 20). Moreover, with the robotic system, the data acquisition can be repeated multiple times at each increment to enhance the dataset. In this case, regression analysis or curve-fitting techniques can be employed to estimate the coefficients A, B, and C based on the extended dataset. The curve-fitting approach provides a more accurate and statistically robust estimation of the coefficients. The primary advantage of the robotic phased array with FMC lies in the significantly increased dataset, which improves the accuracy of coefficient estimation. The larger number of data points allows for capturing finer details, reducing the impact of noise or outliers. Additionally, the FMC technique enables advanced imaging capabilities to create high-resolution stress distribution maps, providing valuable insights into the behaviour of the bolt.

Considering an example and examining actual figures, an initial 20-element array was employed; however, due to the influence of defects, only 12 elements were found to be usable, rendering twelve data points available for each stress increment. Additionally, the FMC approach provided 12 × 12 or 144 data points, while utilising the robotic system to test each point three times resulted in a dataset of 432 data points (see Figure 13 and Figure 14b). For simplicity, we assume the usage of twelve data points for each stress increment, and Figure 13 illustrates four stress increments. The graph depicted in Figure 13 was generated using MATLAB code to solve Equation (8), which is based on the measurement data including ToF measured using the PAUT system (Figure 4) and average stress calculated using FEA analysis with the load cell as input. Polynomial curve fitting was performed using the MATLAB polyfit function, and the optimised parameters were derived from the coefficients. Once the acoustoelastic coefficients were estimated, Equation (8) was adjusted to account for the stress becoming unknown but calculable if the PAUT system measures the ToF (indicated by the blue line in Figure 13). In the single-element approach, the Newton–Raphson method was employed for each individual dataset, yielding a range of stress values based on the considered dataset. The summarized results are presented in Figure 14 and Table 3.

The results demonstrate that PAUT significantly improves the accuracy of stress measurement compared to the single-element approach, which does not incorporate software adjustment. The PAUT measurement accuracy of 8 MPa, compared to the expected value of 8.35 MPa, was deemed highly acceptable. In contrast, the single-element approach, which considers individual acoustic paths, sometimes resulted in complete data blockage due to defects, leading to unreliable stress readings. The robotic method, with hardware adjustment and a consistent couplant layer, further enhanced the accuracy of stress measurement.

It is important to highlight that the measurement error of this specific PAUT data point, 4%, represents one of the best accuracies achieved. Due to the extensive data collection and the limitations of paper size, it is not possible to report all data points here and thus only one example of the FMC approach is shown in Figure 14b. The average stress-measurement error using the robotic PAUT system was approximately 5% as shown in Figure 14c. In comparison, commercially available stress-measurement systems that rely on the single-element approach claim a 5% measurement error. However, our tests revealed that the single-element approach can fail to provide any data in certain positions where the wave is still influenced by defects. The measurement error range for the single-element approach, as observed in the data point reported in Table 3, ranged from 5% to 200%. Therefore, we strongly advise against relying solely on the single-element approach.

The benefits of the robotic PAUT stress measurement for bolts, in comparison to the single-element approach, can be summarized as follows:(1)Simultaneous defect detection and stress measurement: The PAUT system allowed us to leverage advanced features such as the TFM to accurately locate defects. This precise defect mapping facilitated subsequent hardware and software adjustments.(2)Utilisation of extensive data: The abundant data obtained through PAUT enabled us to solve the nonlinear acoustoelastic equations using numerical methods.(3)Accurate stress measurement in the correct position: With the hardware and software adjustments in place, we achieved an average measurement error of 5%, demonstrating the effectiveness of the robotic PAUT approach. Conversely, the single-element approach exhibited measurement errors ranging from 5% to 200%.

#### 4.3.6. Disadvantages of the Robotic PAUT Approach in Offshore Applications

It should be noted that the utilisation of robotic PAUT poses certain challenges when applied in offshore applications. This section outlines two drawbacks associated with this approach:(a)Costs: The implementation of robotic PAUT requires substantial investments in various aspects, including hardware such as robots, phased array probes, controllers, load cells, and force/torque sensors. Additionally, software resources such as LabView 2023, MATLAB R2023b, and Finite Element (FE) software (Abaqus 2023) are necessary. Moreover, the employment of highly skilled personnel adept in data interpretation, numerical modelling, and coding is vital, as exemplified in this study. The financial outlay for this approach can be up to four times greater than that of a manual system employing a single-element transducer.(b)Deployment: It is crucial to recognise that deploying robots and PAUT systems in offshore facilities presents significant challenges. Maintaining the operational integrity of such sophisticated systems requires dedicated efforts and regular maintenance tasks for turbine operation management.

Nevertheless, despite these challenges, the investment in robotic PAUT is deemed worthwhile due to its considerable advantages. Notably, this approach offers significantly improved accuracy and mitigates data-related issues associated with single-element transducer approaches. The achievement of comprehensive and accurate inspections through such an investment would yield substantial returns. It is also anticipated that with rapid advancements in technology and the capability of robots to operate continuously under harsh conditions, the future requirement for regular maintenance of a robot situated within offshore wind turbines (OWTs) will decrease.

#### 4.3.7. Mobile Robotics, Machine Learning-Enabled Technologies, and Continuous Monitoring

The robotic bolt-testing method introduced in this paper holds the potential for continuous monitoring of bolts in various structures using a combination of fixed-arm robots, as used in this paper, and mobile robots such as crawlers and/or drones. By fully automating inspections and data interpretation through the integration of AI-enabled technologies, which can be monitored from onshore facilities, the need to send operators into harsh marine environments would be eliminated. This development not only enhances human safety but also reduces the occurrence of injuries and fatalities.

Given that the robot enables continuous monitoring of bolts, issues such as the appearance of new defects or loosening of the preload can be linked to the load cycle, indirectly monitoring stress-concentration points in the structure. This allows for the quantification of structural load cycles, providing valuable information for predictive maintenance. The integration of Machine Learning-enabled technologies to bridge hardware and software adjustments (as discussed in Section 4.3.3) with predictive maintenance will consequently result in overall reduced maintenance requirements, thereby addressing deployment challenges.

An exemplary scenario for future application of this system in offshore wind farms is shown in Figure 15. If the PAUT end-effectors are connected to robotic facilities, UAVs, crawlers, and ROVs, a comprehensive internal and external inspection of bolts and welds can be conducted. This will provide information on potential defects and, more importantly, early signs of defect generation through monitoring stress. These data can be transferred to onshore facilities where FE simulation and digital twinning can help analyse the load and then send operational commands, such as adjusting the angle of wind turbine blades, to prevent further damage and reduce stress concentration points.

It should be noted that the robotic PAUT approach is versatile for any bolted structure, and the technique’s adaptability enables preload measurement across applications such as pressure vessels, power plants, and offshore oil/gas assets requiring fatigue life assurance. Having co-developed this work with industry partners in the renewable energy sector, the main focus of this work was on enhancing maintenance practices for offshore wind turbines. Despite its generalizability, focusing on wind turbine bolts is pertinent due to the following reasons:(a)Exposure to corrosive environments necessitates continuous integrity monitoring to mitigate loss of preload from bi-metallic connections. Automated testing enables periodic assurance against corrosion-induced bolt failures.(b)Modular structural design involving numerous bolted flange joints is critical for turbines withstanding fluctuating gravitational and aerodynamic loads over decades. This work facilitates condition monitoring with prognostic abilities for these safety-critical bolts.(c)Commitments to minimize operator time offshore drive innovations for remote asset management. The proposed robotics-enabled solution promises capabilities aligning with this strategic objective.

The connection-integrity challenges in wind towers served as the impetus for pursuing advancements in bolted structural health monitoring. Additionally, as a technology incubated to address those particular demands arising from marine environments and modular structural aspects, utilizing the same case brings out the most coherent representation.

#### 4.3.8. Mitigating Offshore Bolt-Corrosion Challenges with Robotic PAUT

Bolt corrosion is a significant challenge in offshore wind turbines exposed to harsh marine environments. While this study does not explicitly discuss corrosion, the proposed robotic PAUT method can detect and account for corrosion defects during bolt testing. The key advantage of PAUT over single-element ultrasonic testing is its simultaneous defect detection and stress-measurement capabilities. Using advanced techniques like TFM, even minor corrosion defects can be precisely mapped in 3D to identify healthy acoustic paths for accurate stress measurement (as shown in Section 4.3.2 and Section 4.3.3). The robot’s position can be optimized through hardware adjustments to avoid corrosion defects obstructing wave propagation. Additionally, software adjustments allow the selection of only healthy A-scans free from corrosion-blocking backwall echoes. Thereby, the impact of corrosion on stress measurement is minimized. The consistent couplant layer and probe pressure ensured by robotics also improve reliability compared to manual inspection. The robotic approach can accurately replicate bolt positions and orientations between laboratory calibrations and in situ testing, providing corrosion compensation. Thus, the proposed methodology can detect corrosion defects, account for them during stress measurement, and enable condition monitoring—addressing critical challenges around bolt corrosion in offshore wind turbines. The continuous monitoring abilities using robotics and automation will allow early identification of corrosion and preload changes before critical failure. Thereby, the findings of this study can significantly enhance structural integrity assessments and predictive maintenance for bolts in offshore wind turbines, considering corrosion concerns.

It should also be noted that marine inspection, where products can have a high level of corrosion, can be challenging due to increased noise in ultrasonic inspection. In this paper, a differentiation is made between noise in two areas: (I) defect detection and (II) preload measurement. For the former, PCI (an in-house MATLAB-programmed code) is utilized to reduce noise, with the details explained in Section 4.3.2. For the latter, the post-processing code for Time-of-Flight (ToF) measurement can estimate a time window where the amplitude peak (representing the backwall echo) is expected. Therefore, noise around the peak is typically not a significant issue as long as noise levels remain normal. It is noted that a low Signal-to-Noise Ratio (SNR) indicates a problematic scan, where either there is a defect in the acoustic path (not a healthy A-scan) or there is a problem in the signal. To improve the signal, a high-pitch array (1.2 mm), as described in Section 3.3, is used, which is typically employed for deep penetration applications.

#### 4.3.9. Alternative Approaches to Robotic PAUT

Apart from the ultrasonic method using the single-element transducer, which was the main approach that was compared with the robotic PAUT proposed in this paper, it is necessary to discuss other approaches as well. Conventional preload measurement solutions like torque wrenches, tension calibrators, and strain gauges come with limitations—discrete monitoring, operator dependencies, and extensive wiring requirements. Modern options aim to address these through convenience and connectivity.

In addition to ultrasonic techniques, other promising methods like Electromechanical Impedance (EMI), coda wave interferometry, and pitch–catch active sensing have emerged for bolt preload monitoring. EMI involves analysing impedance signatures from piezoelectric transducers to detect preload variations through the corresponding mechanical impedance changes they produce in the structure. Analytical modelling and experiments have shown EMI’s ability to correlate impedance metrics to clamping forces [30]. Meanwhile, coda wave analysis utilizes scattering ultrasonic waves in structural materials for sensitivity to minor velocity fluctuations induced by preload loosening. Signal distortions caused by loose bolts have also been characterized effectively using coda wave techniques [31]. Pitch–catch active sensing utilizes pairs of surface-bonded transducers to detect bolt loosening through time shifts in propagating ultrasonic waves when clamping forces are lost. This method has shown promise for structural health monitoring in aircraft bolted joints from tests on representative fuselage structures [32]. These emerging techniques offer complementary strengths—EMI provides high-frequency information, coda waves give scattering insights, and active sensing enables convenient monitoring. Integrating their capabilities can overcome individual limitations to give comprehensive bolt-loading assessments validated by the presented PAUT approach.

The proposed robotic phased-array approach in this landscape signifies a transformative strategy optimizing reliability, automation, and insight richness. PAUT combines extensive penetration for volumetric inspections with precise imaging to visualize subsurface conditions. Robotic deployment eliminates inconsistencies, simultaneously enabling continuous monitoring. Integrated temperature data facilitate real-time compensation while recorded positions/orientations allow replication of conditions between the calibration and installed state. Backed by validated simulation models, the methodology provides a holistic bolt asset-management solution tailored for asset-integrity management in structural connections.

In essence, the presented solution signifies the next generation of preload measurement technology expected in Industry 4.0 environments where offline assays become superseded by online monitoring supported by analytics for predictive maintenance. It is believed that standardizing this methodology in structural-life management protocols can bring unprecedented benefits regarding operational risk reduction.

## 5. Conclusions

This study unveiled a pioneering approach for precise preload measurement in offshore wind bolted connections. The strategic integration of robotic PAUT, nonlinear acoustoelasticity, and FEA yielded a refined methodology, seamlessly harmonising defect detection and stress measurement. Notably, this innovative approach incorporated nonlinearity, meticulously considered average preload, and achieved significantly enhanced measurement accuracy. This cumulative innovation propelled preload measurement techniques forward, underscoring the assurance of structural integrity and operational efficiency in offshore wind systems. The following key conclusions were drawn regarding the robotic PAUT testing of an M36 bolt:Comprehensive and accurate defect detection is critical prior to preload measurement to ensure reliable results. Acceptable defects, smaller than the defined criteria, impact stress measurement, while bolts with significant defects are rejected, obviating the need for further stress measurement.Robotic preload measurement ensures consistent probe pressure and uniform couplant-layer thickness, and maintains consistency between calibration and in situ stress measurement regarding position and orientation. The study demonstrated that a change in orientation can lead to up to 140 MPa error in bolt stress measurement.Considering the average stress is vital for comparison with ultrasonic data. FEA can be employed to provide such information.The advantages of the robotic PAUT method over the single-element approach were discussed. These advantages include incorporating nonlinearity into the equations, simultaneous defect detection and stress measurement, hardware and software adjustments, and, most importantly, a substantial improvement in measurement accuracy.

Based on the results of this paper and considering the established use of PAUT systems for defect detection, the authors highly recommend adopting the robotic PAUT approach for preload measurement. 

## Figures and Tables

**Figure 1 sensors-24-01421-f001:**
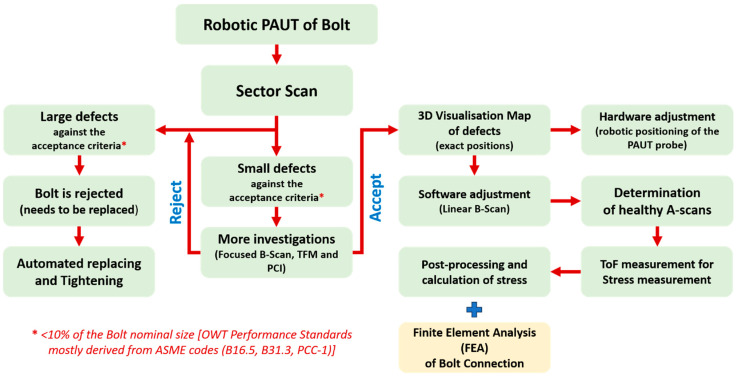
Methodology of the robotic PAUT of the bolt.

**Figure 2 sensors-24-01421-f002:**
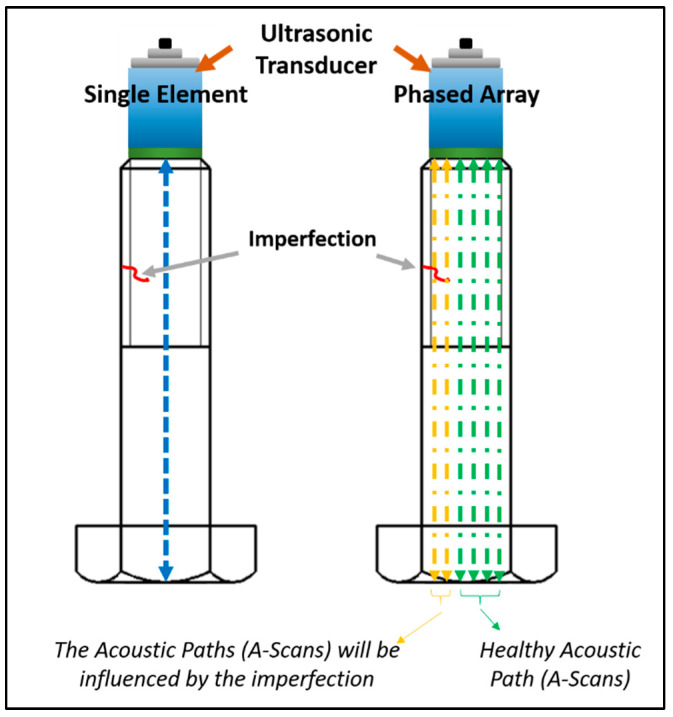
Definition of healthy A-scans in the application of PAUT (instead of the single-element approach) for the bolt preload measurement.

**Figure 3 sensors-24-01421-f003:**
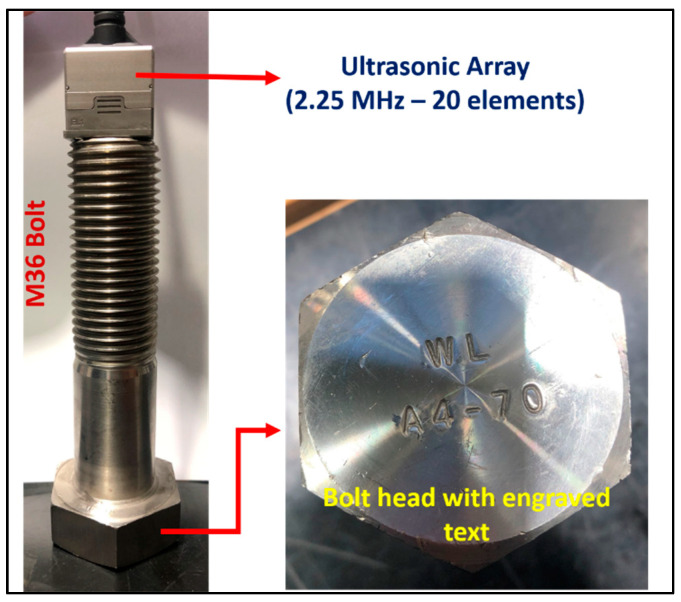
The effect of small defects on the stress.

**Figure 4 sensors-24-01421-f004:**
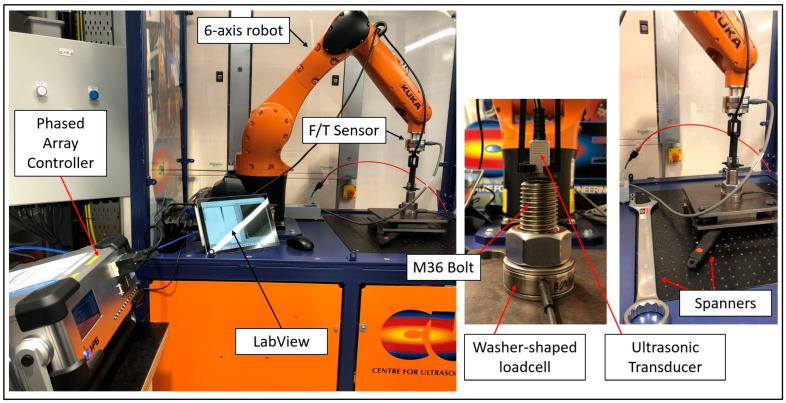
Experimental setup for the robotic PAUT of the bolt.

**Figure 5 sensors-24-01421-f005:**
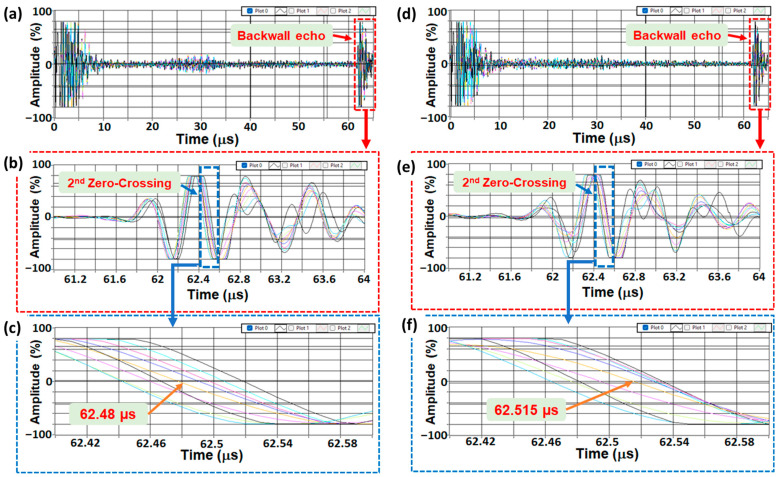
The influence of small defects on the stress: (**a**–**c**) Test#1 and (**d**–**f**) Test#2. The only difference between (**a**), (**b**), and (**c**) is the zoom scale, which allows for the display of all A-Scans (each A-Scan is specified by a different color individually) produced by all elements of the array. Similarly, for (**d**), (**e**), and (**f**).

**Figure 6 sensors-24-01421-f006:**
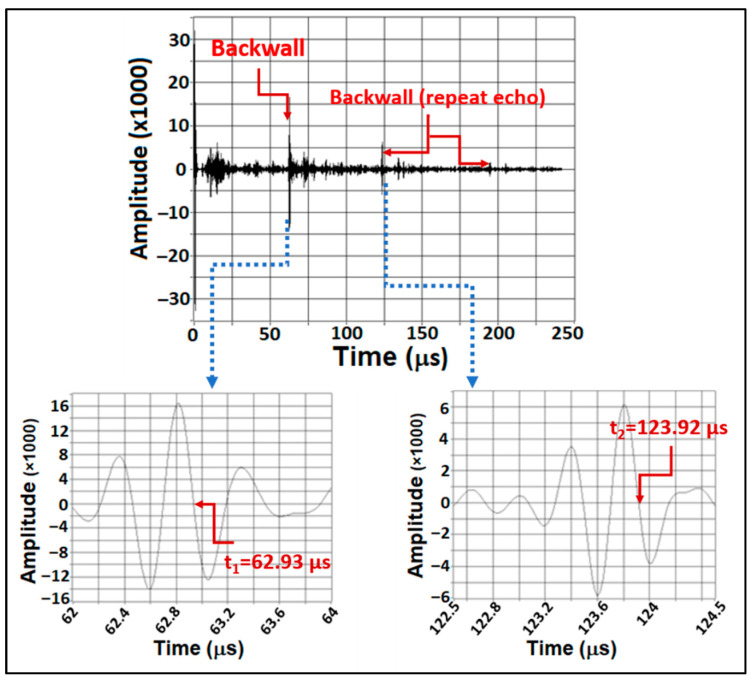
The 1st and 2nd backwall echo.

**Figure 7 sensors-24-01421-f007:**
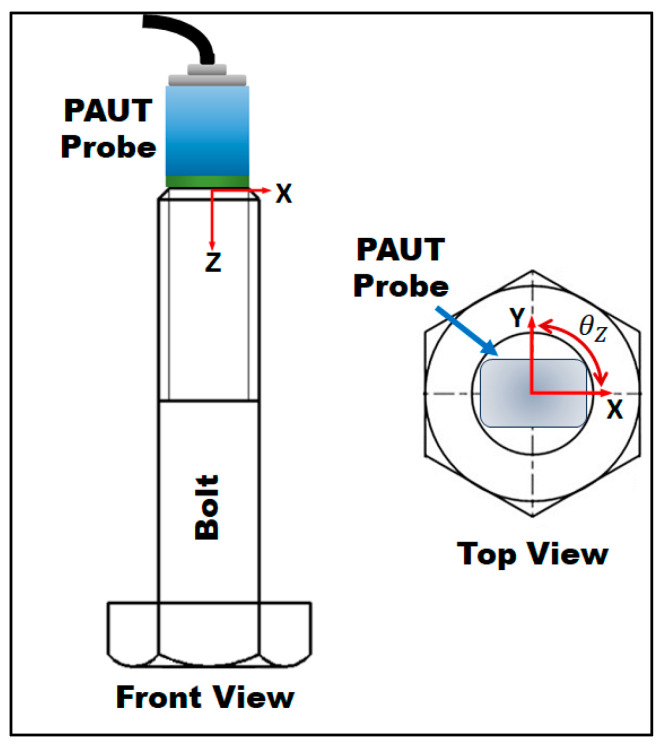
Orientation change for bolt testing.

**Figure 8 sensors-24-01421-f008:**
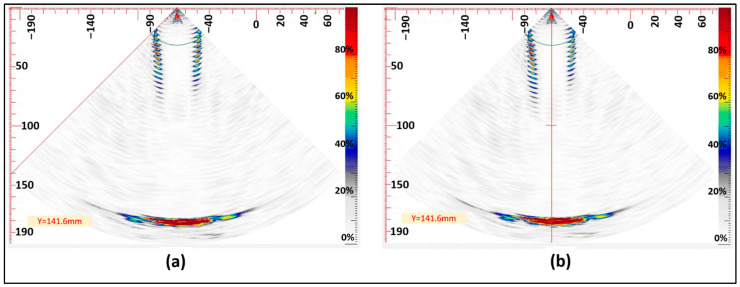
Sector scan of the bolt: θz = 0° (**a**) and θz = 90° (**b**).

**Figure 9 sensors-24-01421-f009:**
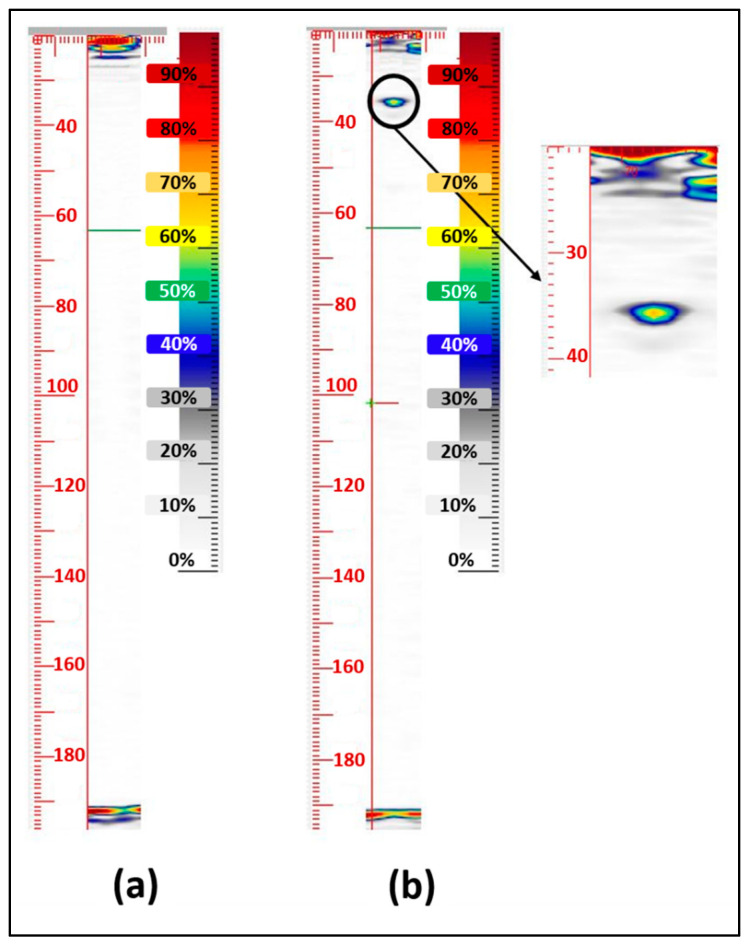
TFM image used to find the defect ((**a**): no defect detected, (**b**): SDH was detected).

**Figure 10 sensors-24-01421-f010:**
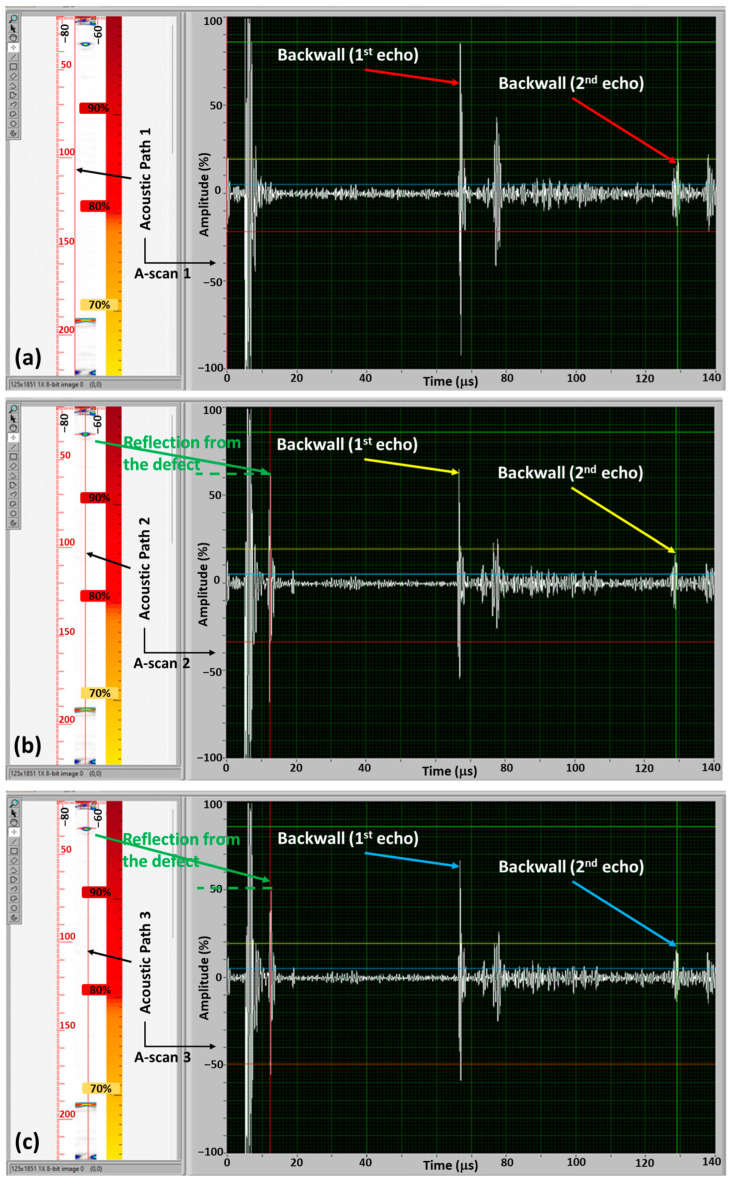
Software adjustment—(**a**) Acoustic Path 1, which excludes the defect, (**b**) Acoustic Path 2, which passes through the centre of the defect and (**c**) Acoustic Path 3, which passes along the edge of the defect.

**Figure 11 sensors-24-01421-f011:**
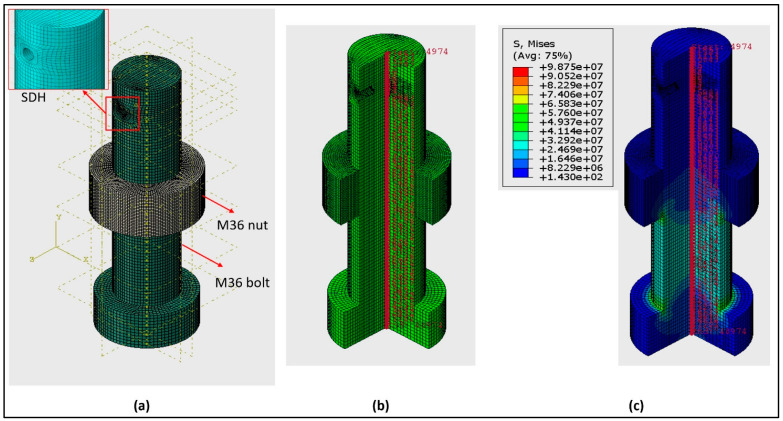
FEA Details: (**a**) Mesh model including the simulated stress concentration defect (SDH), (**b**) the centre path of interest within the model and (**c**) an example of the FEA results for a load of 30 KN.

**Figure 12 sensors-24-01421-f012:**
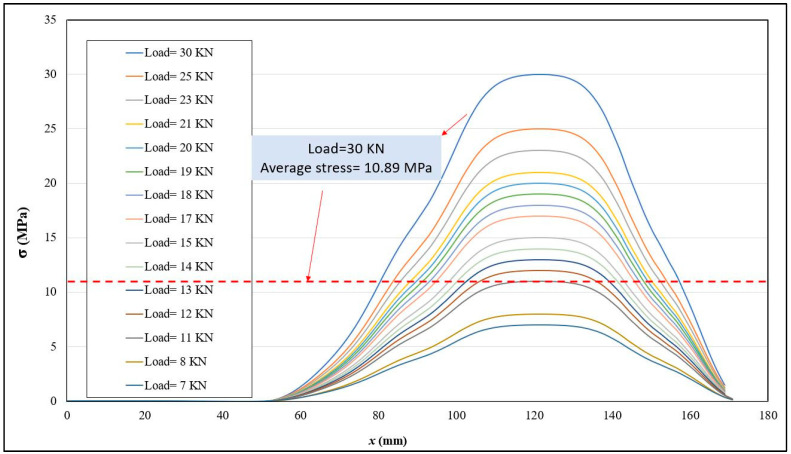
Average stress calculated by FEA to be comparable with the ultrasonic results.

**Figure 13 sensors-24-01421-f013:**
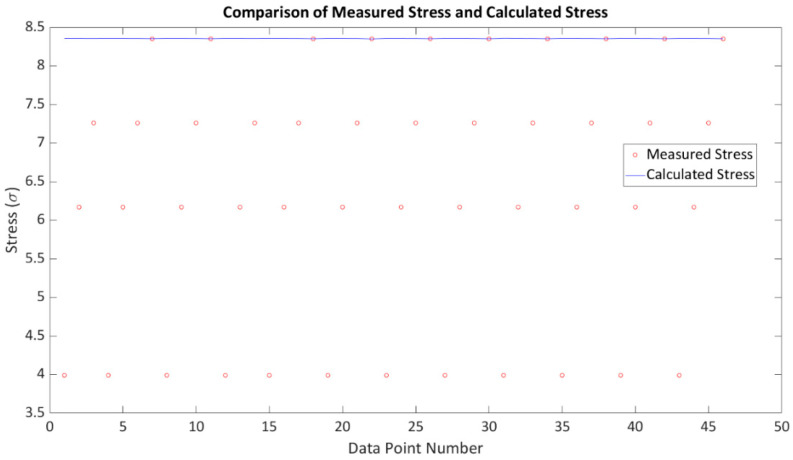
Example of applying a numerical method to solve the acoustoelastic equation using the direct PAUT approach.

**Figure 14 sensors-24-01421-f014:**
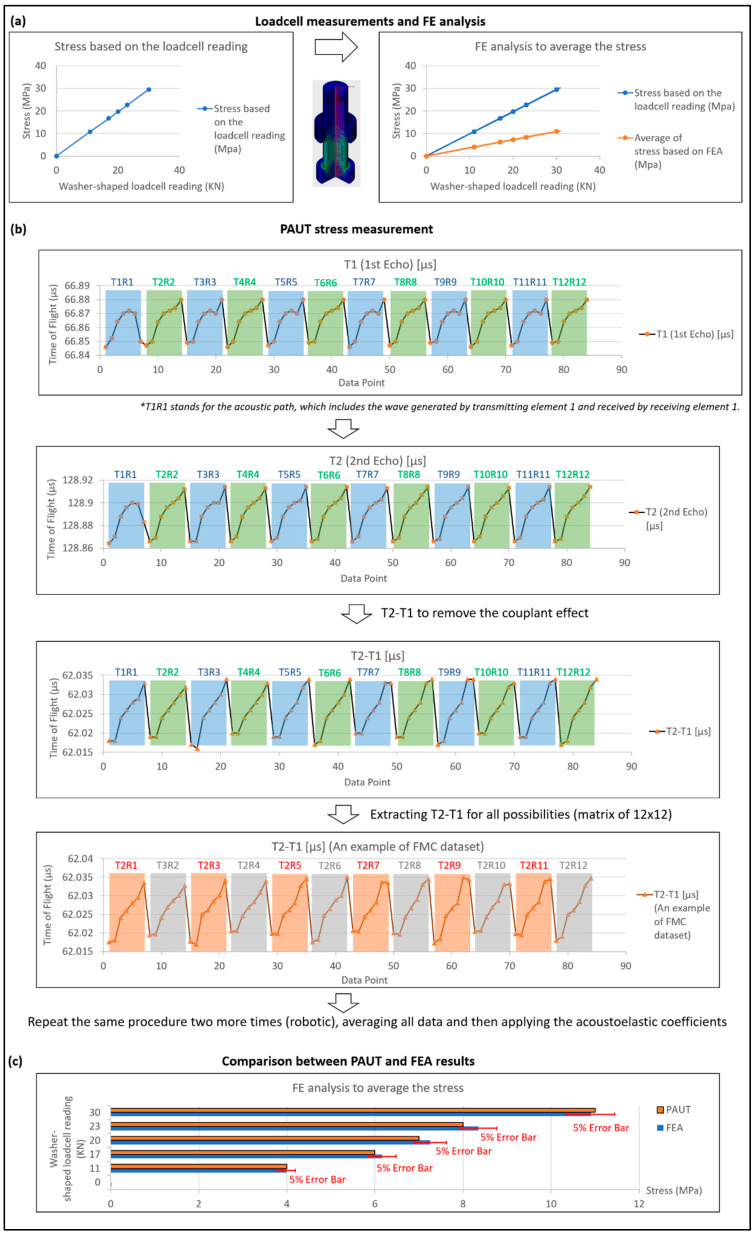
Loadcell measurement and FEA (**a**), PAUT stress measurement (**b**) and comparison between PAUT and FEA results (**c**). * T1R1 stands for the acoustic path, which includes the wave generated by transmitting element 1 and received by receiving element 1.

**Figure 15 sensors-24-01421-f015:**
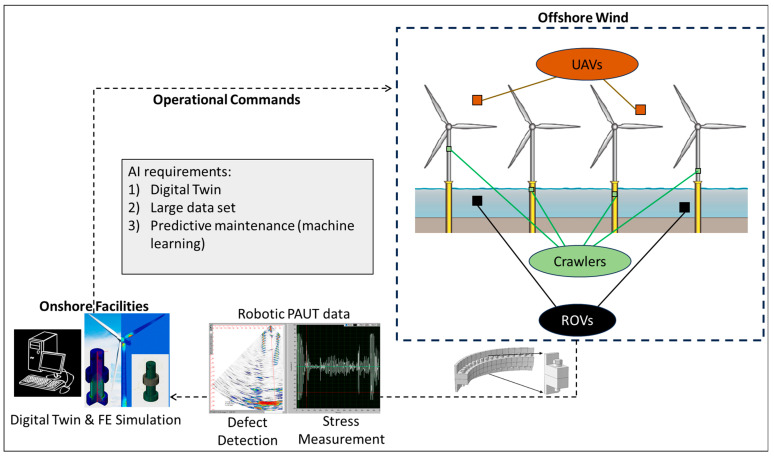
The connection of AI and robotic PAUT for offshore wind applications.

**Table 1 sensors-24-01421-t001:** The difference between the 1st and 2nd backwall echo in Test #1 and Test #2.

	Test #1 (µs)	Test #2 (µs)
t1	62.93	62.91
t2	123.92	123.91
Difference (Δ*t*)	60.99	61.00

**Table 2 sensors-24-01421-t002:** The benefit of robotic testing for PAUT stress measurement.

	Test #A (µs)	Test #B: Different Pressure (µs)	Test #C: Different Orientation (µs)
t1	62.91	62.91	62.88
t2	123.91	123.91	123.86
Difference (Δ*t*)	61.00	61.00	60.98

**Table 3 sensors-24-01421-t003:** Example of stress measurement using PAUT and single-element approach.

Washer-Shaped Loadcell Reading	Stress Based on the Loadcell Reading	Average of Stress Based on FEA	Stress Measured by the PAUT Method	Stress Measured with the Assumption of Single Element
23 KN	23 MPa	8.35 MPa	8 MPa	4–11 MPa

## Data Availability

The data that support the findings of this study are available on request from the corresponding author, (Y.J.).

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
