# Peer review of "Phased Array Ultrasonic Method for Robotic Preload Measurement in Offshore Wind Turbine Bolted Connections"

_sensors, 2024, doi:10.3390/s24051421_

Round 1
Reviewer 1 Report
Comments and Suggestions for Authors
This paper employs a phased array method to measure the preload of bolts. Numerous literature have been published on using ultrasonic echoes for preload measurements, and there are already existing products and devices in the market. What is the innovation introduced in this paper?
The introduction lacks an overview of using ultrasonics for preload measurements of bolted connections.
The structure of the article is confusing, and at the very least, the Methodology section should be placed before the Experimental Setup section.
How was finite element simulation implemented? The reviewer is puzzled.
In the ultrasound signal (A-scan), how is noise removed?
How can the findings of this study be applied to practical offshore wind turbines? Given the corrosive nature of the marine environment, bolt corrosion is a significant concern that this paper does not mention and address.
Comments on the Quality of English LanguageThis paper employs a phased array method to measure the preload of bolts. Numerous literature have been published on using ultrasonic echoes for preload measurements, and there are already existing products and devices in the market. What is the innovation introduced in this paper?
The introduction lacks an overview of using ultrasonics for preload measurements of bolted connections.
The structure of the article is confusing, and at the very least, the Methodology section should be placed before the Experimental Setup section.
How was finite element simulation implemented? The reviewer is puzzled.
In the ultrasound signal (A-scan), how is noise removed?
How can the findings of this study be applied to practical offshore wind turbines? Given the corrosive nature of the marine environment, bolt corrosion is a significant concern that this paper does not mention and address.
Author Response
Reviewer #1’s Comments
(In the revised manuscript, the yellow highlights address Reviewer #1’s comments)
- This paper employs a phased array method to measure the preload of bolts. Numerous literature have been published on using ultrasonic echoes for preload measurements, and there are already existing products and devices in the market. What is the innovation introduced in this paper?
Response: We sincerely appreciate your concern regarding the novelty in our paper, and in response, we have highlighted some discussions to clarify the novelty in Sec. 1 and Sec. 5. You are absolutely right about the availability of numerous literature and even existing products on using ultrasonics for preload measurement. However, to the best of our knowledge, all of them are using single-element ultrasonic transducers, which we have demonstrated in this paper can lead to considerable measurement errors, especially if there is an undetected defect in the bolt. Instead, we have utilized Phased Array Ultrasonic Testing (PAUT) to first detect defects and then conduct the preload measurement only in the healthy acoustic paths. Furthermore, we have integrated both defect detection and preload measurement process, for the first time, with fixed-arm robotics and specifically developed software (LabView and Matlab). There has been no previous publication on the main novelty of our work: using the robotic PAUT method for bolt preload measurement. We believe this paper, if published, will be the inaugural journal paper reporting the outcome of an innovative research project and the first collaboration between the SEARCH robotic lab and the NAOME offshore team (please see the Acknowledgement section of the manuscript for details).
- The introduction lacks an overview of using ultrasonics for preload measurements of bolted connections.
Response: Thank you for bringing this to our attention. We have highlighted the overview of using ultrasonics for preload measurements in Sec. 1, and we have also added a new discussion on using alternative approaches for preload measurement (Sec. 4.3.9, highlighted in green in response to Reviewer #3’s similar comment).
- The structure of the article is confusing, and at the very least, the Methodology section should be placed before the Experimental Setup section.
Response: We appreciate the reviewer’s concern about the structure of the paper. Given the length of the paper and the number of changes needed to address all comments from the four reviewers, it would be challenging to restructure the paper at this stage. However, we have made some minor changes in the title and sub-titles of Sec. 3, where the Methodology (Sec. 3.1) is now placed before the main experimental setup (Sec. 3.3). We have also provided a flowchart (Figure 1) and explanations for almost all parts, except the section on Large Defects, which we clearly stated is beyond the scope of this paper (Sec. 3.1). The structure of Sec. 4.3 is specifically designed to include sub-sections similar to the titles phrased in the methodology figure (Figure 1). For instance, Sec. 4.3.1 covers Sector Scan, Sec. 4.3.2 delves into Further Investigations (focused B-scan, TFM, and PCI), etc. Additionally, Figure 2 has been introduced to elucidate the flowchart and expound upon the definition of 'healthy A-scans.' This graphical representation further enhances the clarity and understanding of the methodology. We hope these efforts help the readers understand the fairly sophisticated system we have used, and we apologize again that we were unable to restructure the paper to fully address this comment.
- How was finite element simulation implemented? The reviewer is puzzled.
Response: Thank you for your comment, and we apologise if the implementation of the FEA was not clarified in the previous version. In the revised manuscript, we have edited some parts in Sec. 3.1 and Sec. 4.3.4 to clarify the usage of FEA in this paper. Here is also our response to this comment, and we have reviewed the manuscript to ensure that these clarifications have been reflected in the revised version now.
- The finite element simulation was conducted to facilitate a meaningful comparison between the stress measurements obtained through our ultrasonic method and those recorded by a washer-shaped load cell in the experimental setup.
- Modelling Approach: The bolt and nut were represented as 3D solids using C3D8R elements for meshing. It is noteworthy that, for the purpose of computational modelling focused on obtaining average stress along the central path and simplifying the simulation, we chose to neglect the impact of the threads.
- Boundary Conditions: Fixed boundary conditions were applied to restrict all degrees of freedom of the contact surfaces of the bolt head and nut in contact with the plate and washer. Additionally, tie contact was established between the bolt and nut contact surfaces, mirroring the physical arrangement in the experimental setup.
- Load Application: A bolt load equal to the data recorded by the washer-shaped load cell was applied to the bolt to generate stress along the central path.
- Intentional Defect: The provided defect in the sample is also modelled in the FE simulation to assess its impact on stress analysis.
- Results Presentation: The FEA mesh and corresponding results, including stress distributions and average stress along the central path, are presented in Figures 11 and 12 of the manuscript.
- This analysis approach enabled determining the stress distribution's practical impact - predictions showed high-stress concentrations near the bolt shank under the nut with gradients lowering towards the free length. Therefore, ultrasonic averaging effects along the entire sound path could be evaluated. Postprocessing the simulations provided the stress profile's mean value along the ultrasonic propagation path inside the bolt, allowing direct comparison against experiments. Thereby, the model complemented physical measurements, considering real-world factors like loss of preload beyond the nut.
- In the ultrasound signal (A-scan), how is noise removed?
Response: Thank you for this comment. We have added a new section (Sec. 4.3.8) to address your next brilliant comment, and the last paragraph of that section discusses the noise to address this comment.
- How can the findings of this study be applied to practical offshore wind turbines? Given the corrosive nature of the marine environment, bolt corrosion is a significant concern that this paper does not mention and address.
Response: Thanks for the constructive comment. The importance of this comment prompted us to add a new section to the revised manuscript (Sec. 4.3.8) to explicitly discuss this.

Reviewer 2 Report
Comments and Suggestions for Authors
I thank you for this article that presents a robotic PAUT approach for defect detection and stress measurement in offshore wind turbine bolted connections. The authors demonstrated the use of robot-enabled PAUT and nonlinear acoustoelasticity measurements over conventional single transducer manual measurements to improve reliability and accuracy. While there’s no novelty in the techniques used in this paper, the authors have delineated a new NDT methodology beneficial for the OWT application. The experimental procedure lacks some important technical aspects that are necessary for the article to be published. These improvements and some other minor corrections are presented below:
1. Section 3.2: Tests 1 and 2 were conducted with and without ultrasonic gel filled in the engraved marks on the bolt head. The presence of ultrasonic gel will moderately reduce the acoustic impedance mismatch and cause a slightly weaker backwall reflection, but the time of flight will remain unchanged. Can you elaborate on how test 1 and 2 conditions are affecting the time of flight measurements?
2. Table 1 and 2: It is standard practice to conduct multiple experimental trials, especially for manual measurements. One trial of TOF measurement is not sufficient.
Author Response
Reviewer #2’s Comments
(In the revised manuscript, the blue highlights address Reviewer #2 ’s comments)
- I thank you for this article that presents a robotic PAUT approach for defect detection and stress measurement in offshore wind turbine bolted connections. The authors demonstrated the use of robot-enabled PAUT and nonlinear acoustoelasticity measurements over conventional single transducer manual measurements to improve reliability and accuracy. While there’s no novelty in the techniques used in this paper, the authors have delineated a new NDT methodology beneficial for the OWT application.
Response: We sincerely appreciate your comment. In regard to the novelty, we acknowledge that we have utilized two well-established concepts (PAUT and robotics) to enhance measurement accuracy. However, we would also like to emphasize that utilizing PAUT for stress measurement is quite novel. It was first introduced for residual stress measurement by the lead author of this paper (Javadi et al, Feasibility study of residual stress measurement using phased array ultrasonic method, ICRS 11, 2022). Additionally, the novelty of the paper is highlighted in yellow in the revised version in response to Reviewer #1's comment. Thank you once again for your valuable input.
- Section 3.2: Tests 1 and 2 were conducted with and without ultrasonic gel filled in the engraved marks on the bolt head. The presence of ultrasonic gel will moderately reduce the acoustic impedance mismatch and cause a slightly weaker backwall reflection, but the time of flight will remain unchanged. Can you elaborate on how test 1 and 2 conditions are affecting the time of flight measurements?
Response: Thank you for your valuable comment. The bolt head was placed on the lab table and even if the ultrasonic wave can propagate in the gel, it will ultimately reflect back from the next material. So, you are correct that the gel can moderately affect the acoustic impedance mismatch, but it can also affect the Time of Flight (ToF) as the acoustic path distance is slightly different. The whole idea of this paper is to demonstrate that what might be considered as a "slight change" in the area of defect detection can be interpreted as a "considerable change" in the area of stress/preload measurement, since the ratio of ToF change to stress change is very small. Therefore, even a few microseconds can result in a measurement error of hundreds of megapascals in stress. In Sec. 3.2, we attempted to show that if some ultrasonic gel can cause a considerable error in stress measurement, this will need to be considered more carefully when the ultrasonic method is suggested for preload measurement in the field, where corrosion, oil/grease, and other external substrates are very common to cover the bolt.
- Table 1 and 2: It is standard practice to conduct multiple experimental trials, especially for manual measurements. One trial of TOF measurement is not sufficient.
Response: We appreciate the reviewer's concern, and you are absolutely right that one trial of ToF measurement is not sufficient. Please note that both Table 1 and Table 2 are generated based on multiple measurements, and only some exemplar results are presented. However, none of those experiments are the main focus of this paper, which was designed based on the robotic PAUT. The purpose of those experiments was to provide counterexamples to demonstrate the level of error we should expect if we don’t consider the possibility of the existence of defects and also rely solely on manual inspection. For the main experiment of the paper, the PAUT data collection enables FMC and consists of hundreds of datasets (e.g., 432 data points, as highlighted in Sec. 4.3.5). Due to the vast amount of data, we had to report only a portion of the tests for simplicity. It was not possible to include all the data in one paper, as highlighted in Sec. 4.3.5 (please also refer to Figure 13 & 14).

Reviewer 3 Report
Comments and Suggestions for Authors
This paper presents an approach for preload measurement of bolted connections. This method combines robotics, PAUT, nonlinear acoustoelasticity, and FEA. Some comments are listed as follows:
(1) As the authors mentioned, there are some applications in PAUT systems to measure stress (mainly residual stress), like welding. What's the main contribution of PAUT in this research?
(2) What is the highest accuracy of the proposed metho?
(3) There are also some other methods related to preload monitoring, for example EMI, coda wave et al. The authors should review more papers. 10.1177/1475921707081979;10.1177/14759217211063420; 10.12783/shm2019/32435
Comments on the Quality of English LanguageMinor editing of English language required
Author Response
Reviewer #3’s Comments
(In the revised manuscript, the blue highlights address Reviewer #3 ’s comments)
- As the authors mentioned, there are some applications in PAUT systems to measure stress (mainly residual stress), like welding. What's the main contribution of PAUT in this research?
Response: Thank you for your comment. Regarding the application of PAUT to measure stress, we would like to highlight that the lead author has actually invented the PAUT stress measurement technique. However, while we have presented this approach at various conferences, no journal paper detailing this method has been published yet. Therefore, if accepted, this paper would be the first journal paper to present this innovative idea. Additionally, the application of PAUT for preload measurement in bolts has substantial differences from its application for residual stress measurement, further emphasizing the novelty of this paper. These differences include:
- Residual stress measurement is typically based on using the Longitudinal Critically Refracted (LCR) ultrasonic wave, while PAUT preload measurement of bolts is based on the pulse-echo configuration.
- The PAUT used for residual stress measurement is not linked to defect detection, while the linkage between defect detection and preload measurement of bolts is the main focus of this paper.
Additionally, the novelty of the paper is highlighted in yellow in the revised version in response to Reviewer #1's comment. Thank you once again for your valuable input.
- What is the highest accuracy of the proposed method?
Response: We appreciate the reviewer's comment. In response, we have highlighted the discussions around accuracy in Sec. 4.3.5 and the error bars are also shown in Figure 14c.
- There are also some other methods related to preload monitoring, for example EMI, coda wave et al. The authors should review more papers. 10.1177/1475921707081979;10.1177/14759217211063420; 10.12783/shm2019/32435
Response: The reviewer made an excellent point regarding reviewing alternative preload monitoring approaches. Thanks for the constructive comment, which drew our attention to the fact that the revised manuscript needed a new section to discuss this. We have then added Sec. 4.3.9 using the helpful papers (the papers are now referenced) the reviewers mentioned.
- Minor editing of English language required
Response: We have conducted an English proofread with the help of AI and also our English-speaking co-authors.

Reviewer 4 Report
Comments and Suggestions for Authors
Please see the attachment.

Author Response
Reviewer #4’s Comments
(In the revised manuscript, the blue highlights address Reviewer #4 ’s comments)
- This manuscript presented a robotic phased array ultrasonic method for preload measurement of bolted connections. The topic is interesting, and the manuscript is overall well-written. I only have a few concerns, as follows, which should be addressed before it can be published.
Response: We really appreciate this positive comment.
- The entire paper did not show any wind turbine structures, but the title of this paper is related to this kind of structure. It seems that the proposed method can be performed for any structures where the bolted connections are needed. Therefore, is it necessary to specify wind turbine as the target structure in this paper?
Response: We sincerely appreciate the reviewer highlighting that the robotic PAUT approach is versatile for any bolted structure. You are absolutely right that the technique's adaptability enables preload measurement across many applications. This brilliant comment encouraged us to add two main concepts to our paper: (I) showing some pictures and discussions on how our system will be used in offshore wind applications and (II) explaining the reason we have focused on offshore wind while discussing the other applications and potential usage of our method. Both of these new aspects are discussed in Sec. 4.3.7 of the revised manuscript.
- The authors may consider marking out the different reflections from the defect in Figure 10. In addition, will the following condition happen if the reflections are mingled with the echo? Or the second echo are hard to find whose amplitude are rather small compared to the reflections. Can the authors add a discussion with this regard?
Response: Thank you for your comment. We have modified Figure 10 to highlight the reflection from the defect. Additionally, we have addressed the issue of defects' reflections being mingled with the backwall echo by adding a new paragraph at the end of Sec. 4.3.3 to provide clarification.

Round 2
Reviewer 1 Report
Comments and Suggestions for Authors
I have no other comments.
Comments on the Quality of English LanguageI have no other comments.